# Is the Integration between Corn and Grass under Different Sowing Modalities a Viable Alternative for Silage?

**DOI:** 10.3390/ani13030425

**Published:** 2023-01-26

**Authors:** Dayenne Mariane Herrera, Wender Mateus Peixoto, Joadil Gonçalves de Abreu, Rafael Henrique Pereira dos Reis, Fabiano Gama de Sousa, Ernando Balbinot, Vanderley Antônio Chorobura Klein, Ricardo Pereira Costa

**Affiliations:** 1Graduate Program in Tropical Agriculture, Faculty of Agronomy and Animal Sciences, Federal University of Mato Grosso, Cuiabá 78060-900, MT, Brazil; 2Federal Institute of Education, Science and Technology of Rondônia, Colorado do Oeste 76993-000, RO, Brazil; 3Federal Institute of Education, Science and Technology of Tocantins, Pedro Afonso 77710-000, TO, Brazil

**Keywords:** crop–livestock integration, delayed sowing, *Urochloa brizantha*

## Abstract

**Simple Summary:**

Silage production to meet the nutritional demands of ruminant animals in times of forage shortage may present certain limitations, and the adoption of production strategies associated with environmental conservation is of great importance for improving resource utilization. Corn silage is one of the most important components in ruminant feed, since its plant provides a large volume of palatable, highly digestible, and energy-rich feed, which has excellent potential for animal production. In addition, crop–livestock integration also presents itself as an attractive strategy for being promising yield alternatives in the global reality of agriculture production and cattle-raising. The strategy of intercropping is oriented towards silage of better nutritional quality, in addition to helping to maintain sustainable agroecosystems and providing benefits in economic, environmental, and social aspects. Thus, this experiment found that corn silage intercropped with tropical forages in different sowing modalities can be recommended as appropriate for the production of silage to supply livestock with feed, allowing a sustainable intensification of the production system.

**Abstract:**

This study aimed to evaluate the fermentation pattern and dry-matter losses in corn (*Zea mays* L.) silage intercropped with *Urochloa brizantha* cv. Marandu and *Megathyrsus maximus* cv. Mombasa grasses in different sowing modalities through crop–livestock integration. The experimental design was in randomized blocks, which were arranged in a 2 × 5 factorial scheme with four repetitions. The first factor consisted of the grass cultivars Marandu and Mombasa. The second factor was the sowing modalities of grasses intercropped with corn: (1) simultaneous row sowing and inter-row corn sowing (no fertilizer); (2) simultaneous row sowing and inter-row corn sowing (with fertilizer); (3) simultaneous sowing with double grass row in the corn inter-row; (4) delayed sowing inter-row at 7 days after corn emergence; and (5) delayed sowing inter-row at 14 days after corn emergence. The forage buffer capacity (BC), silage pH and ammoniacal nitrogen (NH_3_-N) content, forage (FORDM) and silage dry-matter (SILDM) percentages, gas losses (GL), effluent losses (EL), and dry-matter recovery (DMR) parameters on the ensilage were evaluated. Only forage BC, silage NH_3_-N, and silage DMR variables differed (*p* < 0.05) from the control silage (monocropped corn) when the integration was carried out. The grass cultivar factors and sowing modalities for BC and NH_3_-N variables had an effect. The intercropping of corn and Marandu grass or Mombasa grass, in any grass sowing modality, did not affect the quality of the silage.

## 1. Introduction

Food safety and environmental responsibility have gained importance in decision making on the purchase of animal products. Therefore, the target of Brazilian agriculture should be the search for efficient production systems with flexibility to adapt to these demands. Crop–livestock integration, in this context, stands out as a modern and conservationist technology [1,2].

From the livestock perspective, there are several obstacles to the productivity of the systems, such as pasture degradation, the shortage of roughage, and the loss of nutritional value of feeds for cattle during the dry season of the year [3]. Primary factors that are responsible for pasture degradation include the choice of forage species, the non-replacement of nutrients exported during the grazing season, and incorrect pasture management due to overgrazing [4].

In this regard, the cattle rancher is faced with a double problem: the need to recover pasture productivity and produce and store good quality roughage for the most critical time of the year—the dry season. To this end, growing corn (*Zea mays* L.) intercropped with a tropical forage appears as a solution to both problems [5,6], characterizing the crop–livestock integration system, with emphasis on pasture renewal [7].

The intercropped systems of forage and corn, in many respects, do not reduce grain or mass productivity relative to monocrop corn [8,9]. Progressing towards the objective of supplying quality forage to the animals, especially during the dry season, the forage conservation through ensilage, singularly when coming from intercropped crops, presents itself as an important alternative to meet such demand [10]. Contrastingly, special attention must be paid to the fermentation pattern of the ensiled material. Such attention is based on the fact that forage grasses fermentation has some obstacles that directly interfere with the rapid pH reduction [11]. This does not happen with the corn crop, which is considered standard for silage, due to its fermentative characteristics [12].

For this reason, it becomes important to understand the silage produced via intercropping, since the forage harvester’s cutting platform not only collects corn plants but also the grass that is above the harvest height. It is also necessary to consider that the amount of grass mass in the system is dependent on its sowing modality in relation to the corn, either simultaneously or in a delayed way. Therefore, this study aimed to evaluate the fermentation pattern and dry-matter losses in corn silage intercropped with *Urochloa brizantha* cv. Marandu and *Megathyrsus maximus* cv. Mombasa grasses in different sowing modalities through crop–livestock integration.

## 2. Materials and Methods

### 2.1. Experimental Site

The experiment was carried out in the vegetable production area of Federal Institute of Education, Science and Technology of Rondônia (IFRO), located in Colorado do Oeste, RO, Brazil (13°07′39” S; 60°29′68” W; 460 m altitude). The soil is classified as Eutrophic Red Argisol, with clay texture and flat-wavy slope gradient. Soil analysis performed in the 0 to 20 cm layer revealed the following chemical characteristics: pH H_2_O = 6.6; P = 8.2 mg dm^−3^; K = 68.0 mg dm^−3^; OM = 2.6%; Al = 0.0 cmol_c_ dm^−3^; Ca = 7.9 cmol_c_ dm^−3^; Mg = 0.7 cmol_c_ dm^−3^; cation exchange capacity = 12.1 cmol_c_ dm^−3^; and base saturation = 73.1%. The local climate is classified as tropical monsoon (Am) according to the Köppen–Geiger classification, with two well-defined seasons.

### 2.2. Experimental Design and Management

The experimental design was in randomized blocks, which were arranged in a 2 × 5 factorial scheme with four repetitions. The first factor consisted of grass cultivars: Marandu and Mombasa. The second factor was based on the grass sowing modalities: (1) simultaneous row sowing and inter-row corn sowing (SR + IR); (2) simultaneous row sowing and inter-row corn sowing (with fertilizer) (SR + IRF); (3) simultaneous sowing with double grass row in the corn inter-row (S2IR); (4) delayed inter-row sowing at 7 days after corn emergence (7DAE); and (5) delayed inter-row sowing at 14 days after corn emergence (14DAE), totaling 44 experimental units, with 40 plots coming from factor combinations and four from the control treatment (monocropped corn).

Before sowing crops, a previous survey of the weed plant community was carried out, followed by chemical weed control with the systemic herbicides glyphosate (2.0 L ha^−1^) and dimethylamine salt 2,4-dichlorophenoxy acetic acid (0.74 L ha^−1^) at a flow rate of 200 L ha^−1^. After controlling weeds, the soil was prepared with a plow harrow and, subsequently, a leveling harrow. The intercropping was sown on 9 November 2013, seven days after herbicide application.

In the sowing fertilization, the basic recommendation for the corn crop was followed, in doses of 20 kg ha^−1^ of N, 100 kg ha^−1^ of P_2_O_5_, and 60 kg ha^−1^ of K_2_O and, for the grass, additional fertilization of 20 kg ha^−1^ of N, 80 kg ha^−1^ of P_2_O_5_, and 20 kg ha^−1^ of K_2_O, according to specific treatment requirement [13]. At 35 days after sowing, topdressing fertilization was carried out, in doses of 130 kg ha^−1^ of N and 40 kg ha^−1^ of K_2_O. The corn cultivar (BG 7049 Hx, triple hybrid) was sown on the fertilized furrow, in a depth of 3 cm, to obtain 60,000 plants ha^−1^, with 0.90 m of inter-row spacing. The grasses sowing was carried out in the row and inter-row of the corn sowing, at an average depth of 6 to 8 cm below the corn seed, using a sowing rate of 4 kg ha^−1^ of pure live seed, according to the proposed treatments. Cover fertilization and phytosanitary management were carried out according to technical recommendations for growing corn [14].

The experimental unit in the field consisted of five rows of corn plants, spaced 0.90 m apart and 5.0 m long, totaling 22.5 m^2^. The useful area consisted of the three central rows, disregarding 1.0 m at the ends of the rows. Sampling for ensilage was carried out by collecting the material contained in the useful area of the experimental units at 0.20 m above the ground surface, simulating the cut made by the forage harvester. The intercropped material was collected manually, when the corn plants had grains at the farinaceous stage and the forage was crushed in a stationary chopper into 2 to 3 cm particles immediately before ensiling. The chopped material was divided into two portions of quantified mass (Table 1).

### 2.3. Ensiling and Assessment of Silage Samples

Two experiments were carried out: the first to evaluate the fermentative pattern; the second for the evaluation of dry-matter losses in ensiling. In both experiments, the same experimental design was adopted in the field.

The experimental units in the field became experimental silos (glass jars), with a volume of 2.5 L, equipped with a lid for sealing, and adapted with a “siphon” type valve, to allow the exit and prevent the entry of gases inside the silo. The experimental silos were filled with enough forage for their complete filling after compaction, obtaining an average density of 614.4 kg m^−3^ of green mass.

The forage compaction was carried out manually and, after filling, the silos were closed, applying a layer of acetic silicone to the edges of the lids for complete sealing. The “siphon” valves were filled with water. The silos remained stored for 72 days, without direct incidence of light.

### 2.4. Fermentation Pattern in Ensiling

When the silos were opened, samples were collected in the geometric center of the pot, disregarding the upper and lower portions. At the time of ensiling and opening the experimental silos, forage and silage samples were collected, respectively. The samples were stored in plastic bags and frozen to determine the percentage of dry matter (DM) by lyophilization, as well as determining the total nitrogen (TN) content, according to [15]. To determine the percentage of DM in the forage (FORDM) and silage (SILDM) by lyophilization, we weighed approximately 30 g of frozen sample, in triplicate, which was taken to the lyophilizer (LIOTOP; L101) for 72 h until reaching constant mass, the DM being determined by mass difference.

The second part was used to obtain the aqueous extracts to determine the ammoniacal nitrogen (NH_3_-N) content, pH, and buffer capacity (BC). The NH_3_-N content was quantified from the aqueous extract in the proportion of 1:2, distilled with 2N KOH in a micro-Kjeldahl apparatus [16]. The pH analysis was performed using the method proposed by [17] and BC was determined using the methodology indicated by [18].

### 2.5. Dry-Matter Losses in Ensilage

Dry-matter losses in forages in the form of gases (GL) and effluents (EL) were quantified by mass difference. To determine these, sand and plastic screen were added to the experimental silos in order to collect the effluent during the fermentation process. The silo components were weighed before ensiling: glass jar, screwable lid, dry sand, and plastic screen. After filling the silos, they were weighed again. At the time of opening the experimental silos, these were weighed full to determine the gas losses; then, after removing the silage, the silo, lid, and wet sand set was weighed, quantifying the effluent losses.

To determine these losses and dry-matter recovery (DMR), equations adapted from [19] were used, where the gas losses (% of DM) were quantified by the difference in dry forage mass.

### 2.6. Statistical Analysis

The data obtained were submitted to the normality test and all variables, except the GL, followed a normal distribution, performing the transformation of the square-root function (√x). Subsequently, the bilateral Dunnett’s test was performed for comparison with the control treatment; when the F-test was significant, Tukey’s multiple comparison test was performed, all at 5% error probability, from Sisvar software, version 5.6 [20].

## 3. Results

The fermentative pattern and the dry-matter losses of the silage obtained from the intercropping between corn and Marandu and Mombasa grasses using different sowing modalities were analyzed in comparison with the control treatment (monocropped corn), and the results are presented in Table 2. The forage BC, the silage NH_3_-N content, and the DMR in the intercropped silage differed from the control silage.

For the forage BC, higher values were found due to the use of the SR + IR and 14DAE modalities in the corn and Mombasa grass intercropping. The NH_3_-N contents were higher with the simultaneous S2IR modalities, with good grass participation intercropped, and delayed at 7DAE and 14DAE, with younger, wetter grasses and a high crude protein content.

There was an interaction effect between the grasses and sowing modalities factors on the forage buffer capacity (Table 3). Higher values of BC were verified in the intercropping of corn and Mombasa grass in the modalities SR + IR, S2IR, and 14DAE, compared with intercropping with Marandu grass. There was an effect of the sowing modalities on this variable only in the intercropping between corn and Mombasa grass, so that the delayed sowing at 14DAE (2.23 eq. mg. NaOH 100g ^−1^ of DM) was higher than at 7DAE (1.85 eq. mg NaOH 100g ^−1^ of DM).

There was no significant interaction effect, nor the effect of isolated factors, for the variables pH, FORDM, SILDM, GL, EL, and DMR (Table 4).

There was a significant effect of the interaction between grasses and sowing modalities on the NH_3_-N content (Table 5). The NH_3_-N contents were higher in corn silages intercropped with Marandu grass. Among the sowing modalities, the highest NH_3_-N contents were observed for S2IR (3.68%) and 7DAE (3.17%) in corn silages intercropped with Marandu and Mombasa grasses, respectively.

## 4. Discussion

The observations made in this work show that the forage mass presence from grass intercropped with corn does not cause excessive or negative changes in terms of qualitative parameters in ensilage. The grass presence in the integration biomass with the SR + IR modality may have favored the forage buffering effect, whereas within the 14DAE modality, it has a high crude protein content associated with younger age. This factor can also increase forage buffering [21].

As performed by [22], it can be inferred that although the simultaneous growing of corn and grass promotes a longer season of competition between the crops, there is a compensating effect due to the greater amount of dry matter produced, in addition to the maintenance of qualitative parameters evidenced. The other combinations had no effect on this parameter, as verified by [23], in the intercropping between corn and *Urochloa* or *Megathyrsus* grasses.

Differences in NH_3_-N concentration in monocropped and intercropped corn silage are due to the fact that corn has a higher soluble carbohydrates concentration readily available for fermentation [6]. Those authors, evaluating the fermentative parameters of corn silages integrated with Paiguas palisade grass (*U. brizantha*) in different systems and maturity stages in the off-season, reported that, although there were statistical differences, it was verified that the mean values of NH_3_-N in the silages of all integrations remained low, as in the present study.

The NH_3_-N, expressed as a percentage of total nitrogen, is an important property in silage evaluation because it indicates the amount of protein that is degraded during the fermentation stage or the potential occurrence of overheating of the mass in the silo, causing Maillard reactions [24].

An appropriate result from the low mean values of NH_3_-N in the silage is the adequate lactic fermentation, contributing to proteolysis reduction and the undesirable microorganisms inhibition, as pointed out by [25] in their interpretation and use of silage fermentation analysis reports.

Younger, wetter grass with a high crude protein content in the delayed modality at 14 days after emergence has a direct influence with the lower buffer capacity of the integration, a condition also reported by [6]. Nevertheless, although tropical grasses have a high buffer capacity [11], their presence was not enough to raise this variable to values above tolerated levels. The contrary was reported by [5] when evaluating the fermentative characteristics and the quality of sorghum (*Sorghum bicolor* (L.) Moench.) silages and *U. brizantha* cultivars in monocropped and intercropped corn in different sowing systems in the off-season. Such authors reported that there was no difference in the buffer capacity values, and the values they observed were also within what is desirable for this parameter.

The reduction in ensiling losses and the secondary fermentation inhibition are important goals to be achieved; therefore, several researchers recommend a dry-matter content of around 30%, based on the justifications of greater green mass production, harvesting ease, efficiency in the compaction, and greater grains fragmentation [26]. Although forage grasses have a characteristic of low DM compared with the corn [11,27] and, consequently, are a focus of attention in intercropping ensilage, there was no qualitative impediment in the present study, which obtained good pH, DM, and dry-matter losses ratings throughout the process.

Several benefits are generated by the adoption of management techniques of forage species destined to the silage production. These include mixing naturally dried crops with high-moisture forages, which can reduce the concentrations of butyric acid, ammoniacal nitrogen, and pH in the silage [11]. In addition, its productive potential seems to be amplified, and benefits related to lower production costs can be noted, making it possible to explore a wider range of forage species.

When considering the NH_3_-N content, this is an indicator of silage quality and characterizes the fermentative profile [28]. The observed configuration may be related to the influence of the grass mass presence in the ensiled material, according to the sowing modality. The higher NH_3_-N content verified by the 7DAE modality use can be explained by the intercropped grass age and the high fraction of protein and moisture [11,29,30], which are inherent characteristics of younger grasses.

## 5. Conclusions

Corn silage intercropped with Marandu or Mombasa grasses in grass sowing modalities, such as simultaneous row sowing and inter-row corn sowing, simultaneous row sowing and inter-row corn sowing (with fertilizer), simultaneous sowing with a double grass row in the corn inter-row, delayed sowing in the inter-row at 7 days after corn emergence, and delayed sowing in the inter-row at 14 days after corn emergence, presented adequate fermentation patterns.

Simultaneous and delayed grass sowing can be recommended as suitable sowing systems for silage production. Additionally, the silages produced from intercropped systems proved to be an alternative option for supplying roughage, allowing the sustainable intensification of the production system.

## Figures and Tables

**Table 1 animals-13-00425-t001:** Corn (*Zea mays* L.) and grasses green mass yield and the grasses participation in the total biomass of corn and grass intercropping under different sowing modalities.

Treatment	Yield (kg ha^−1^) *		Grass Participation in the Mass (%)
	Corn	Grass	
Monocropped corn	71,185.5	–	–
Marandu, SR + IR	63,195.4	4652.8	7.4
Marandu, SR + IRF	66,000.0	3547.4	5.4
Marandu, S2IR	60,230.3	3416.5	5.7
Marandu, 7DAE	69,566.1	88.1	0.1
Marandu, 14DAE	67,250.3	30.7	0.1
Mombasa, SR + IR	64,583.3	11,750.0	18.2
Mombasa, SR + IRF	62,773.7	15,722.2	18.8
Mombasa, S2IR	63,941.4	8187.5	12.8
Mombasa, 7DAE	67,601.9	345.8	0.5
Mombasa, 14DAE	70,456.8	42.3	0.1

* Values referring to the green mass harvested from the field. SR + IR: simultaneous row sowing and inter-row corn sowing; SR + IRF: simultaneous row sowing and inter-row corn sowing (with fertilizer); S2IR: simultaneous sowing with double grass row in the corn inter-row; 7DAE: delayed sowing inter-row at 7 days after corn emergence; and 14DAE: delayed sowing inter-row at 14 days after corn emergence.

**Table 2 animals-13-00425-t002:** Means of the treatments for the variables buffer capacity (BC), pH, ammoniacal nitrogen as part of the total nitrogen (NH_3_-N/TN) content, percentage of forage (FORDM) and silage dry matter (SILDM), gas losses (GL), effluent losses (EL), and dry-matter recovery (DMR) in corn (*Zea mays* L.) ensilage intercropped with Marandu (*Urochloa brizantha*) and Mombasa (*Megathyrsus maximus*) grasses compared with the monocropped corn (control).

Treatments	BC eq. mg. NaOH 100 g^−1^ of DM	pH	NH_3_-N % of TN	Dry Matter (%)	
				FORDM	SILDM
Monocropped corn	1.81	3.80	2.13	31.39	31.16
Marandu, SR + IR	1.59	3.81	2.61	31.63	28.77
Marandu, SR + IRF	1.77	3.83	2.20	31.63	30.22
Marandu, S2IR	1.81	3.82	3.68 *	31.13	30.26
Marandu, 7DAE	1.87	3.79	3.39 *	32.16	29.57
Marandu, 14DAE	1.82	3.81	3.02 *	31.16	30.27
Mombasa, SR + IR	2.20 *	3.82	1.73	30.82	26.62
Mombasa, SR + IRF	1.86	3.81	2.38	30.85	30.21
Mombasa, S2IR	2.09	3.81	2.45	31.25	30.09
Mombasa, 7DAE	1.85	3.80	3.17 *	30.18	30.57
Mombasa, 14DAE	2.23 *	3.80	2.22	31.20	30.22
HSD ^1^	0.38	0.15	0.50	2.55	0.97
CV ^2^(%)	9.77	2.03	9.40	4.00	5.03
	**Dry-Matter Losses**			**DMR** (%)	
	GL % of DM	EL kg t^−1^ of green matter			
Monocropped corn	2.07	51.04		90.57	
Marandu, SR + IR	1.79	51.95		76.69 *	
Marandu, SR + IRF	1.78	48.71		88.12	
Marandu, S2IR	2.33	34.21		91.57	
Marandu, 7DAE	2.09	57.44		84.25	
Marandu, 14DAE	2.32	34.87		91.17	
Mombasa, SR + IR	2.01	43.82		87.39	
Mombasa, SR + IRF	2.19	35.42		90.08	
Mombasa, S2IR	2.35	49.28		86.82	
Mombasa, 7DAE	2.24	48.35		91.51	
Mombasa, 14DAE	2.26	41.29		90.39	
HSD ^1^	0.91	24.90		11.64	
CV ^2^(%)	20.97	27.01		6.47	

^1^ Honestly significant difference. ^2^ Coefficient of variation (%). Means followed by asterisk (*) in the column differ (*p* < 0.05) in relation to the control treatment (monocropped corn) at 5% probability of error by the Dunnett’s test. SR + IR: simultaneous row sowing and inter-row corn sowing; SR + IRF: simultaneous row sowing and inter-row corn sowing (with fertilizer); S2IR: simultaneous sowing with double grass row in the corn inter-row; 7DAE: delayed sowing inter-row at 7 days after corn emergence; and 14DAE: delayed sowing inter-row at 14 days after corn emergence.

**Table 3 animals-13-00425-t003:** Corn (*Zea mays* L.) forage buffer capacity (eq. mg NaOH 100 g^−1^ of DM) values intercropped with the Marandu (*Urochloa brizantha*) and Mombasa (*Megathyrsus maximus*) grasses in different sowing modalities.

Sowing Modalities	Grass Cultivars Intercropped		Means
	Marandu	Mombasa	
SR + IR	1.59 aB	2.02 abA	1.89
SR + IRF	1.77 aA	1.86 abA	1.82
S2IR	1.81 aB	2.09 abA	1.95
7DAE	1.87 aA	1.85 bA	1.86
14DAE	1.82 aB	2.23 aA	2.03
Means	1.77	2.04	–
CV ^1^(%)	9.49		

^1^ Coefficient of variation (%). Means followed by the same lowercase letter in the column and uppercase in the line do not differ statistically from each other by Tukey’s test at 5% probability. SR + IR: simultaneous row sowing and inter-row corn sowing; SR + IRF: simultaneous row sowing and inter-row corn sowing (with fertilizer); S2IR: simultaneous sowing with double grass row in the corn inter-row; 7DAE: delayed sowing inter-row at 7 days after corn emergence; and 14DAE: delayed sowing inter-row at 14 days after corn emergence.

**Table 4 animals-13-00425-t004:** Values of silage pH, forage dry matter (FORDM), silage dry matter (SILDM), gas losses (GL), effluent losses (EL), and dry-matter recovery (DMR) of corn (*Zea mays* L.) intercropped with Marandu (*Urochloa brizantha*) and Mombasa (*Megathyrsus maximus*) grasses in different sowing modalities.

Variables	pH	Dry Matter (%)		Dry-Matter Losses		DMR (%)
		FORDM	SILDM	GL * (% of DM)	EL (kg t^−1^ of GM **)	
Marandu	3.81	31.54	29.82	2.06	45.44	86.36
Mombasa	3.81	30.86	30.14	2.21	43.63	89.24
**Sowing Modalities**						
SR + IR	3.81	31.23	29.19	1.90	47.89	82.04
SR + IRF	3.82	31.24	30.21	1.98	42.06	89.10
S2IR	3.81	31.19	30.17	2.34	41.75	89.20
7DAE	3.80	31.17	30.07	2.16	52.89	87.88
14DAE	3.81	31.18	30.25	2.29	38.08	90.78
CV ^1^(%)	2.13	3.75	5.24	21.75	25.63	6.60

^1^ Coefficient of variation (%). * Data transformed by √x. ** GM: green mass. SR + IR: simultaneous row sowing and inter-row corn sowing; SR + IRF: simultaneous row sowing and inter-row corn sowing (with fertilizer); S2IR: simultaneous sowing with double grass row in the corn inter-row; 7DAE: delayed sowing inter-row at 7 days after corn emergence; and 14DAE: delayed sowing inter-row at 14 days after corn emergence.

**Table 5 animals-13-00425-t005:** Ammoniacal nitrogen as part of the total nitrogen (% of NH_3_-N/TN) of corn (*Zea mays* L.) silage intercropped with Marandu (*Urochloa brizantha*) and Mombasa (*Megathyrsus maximus*) grasses in different sowing modalities.

Sowing Modalities	Grass Cultivars Intercropped		Means
	Marandu	Mombasa	
SR + IR	2.61 cdA	1.73 cB	2.17
SR + IRF	2.20 dA	2.38 bA	2.29
S2IR	3.68 aA	2.45 bB	3.07
7DAE	3.39 abA	3.17 aA	3.28
14DAE	3.02 bcA	2.22 bcB	2.62
Means	2.98	2.39	–
CV ^1^(%)	9.09		

^1^ Coefficient of variation (%). Means followed by the same lowercase letter in the column and uppercase in the line do not differ statistically from each other by Tukey’s test at 5% probability. SR + IR: simultaneous row sowing and inter-row corn sowing; SR + IRF: simultaneous row sowing and inter-row corn sowing (with fertilizer); S2IR: simultaneous sowing with double grass row in the corn inter-row; 7DAE: delayed sowing inter-row at 7 days after corn emergence; and 14DAE: delayed sowing inter-row at 14 days after corn emergence.

## Data Availability

The data used in this study are available from the corresponding authors on request.

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
