# Peer review of "Is the Integration between Corn and Grass under Different Sowing Modalities a Viable Alternative for Silage?"

_animals, 2023, doi:10.3390/ani13030425_

Round 1

Reviewer 1 Report

The objectives of the experiments are clear with the intention of studying the effects of grass interseeding with corn silage.  The methods and design appear to be adequate for the work.

 Some sentence structure or wording could be improved throughout the manuscript.  For example,

Line 54 - 57,

 Primary factors that are responsible for pasture degradation include choice of forage species, the non-replacement of nutrients exported during the grazing season and the incorrect pasture management due to overgrazing.

 Line 112:  The grass sowing was…; rather than, The grasses sowing…

 What was the days after sowing when harvest occurred?

 Table 1: Are the yield units (kg/ha) correct? Yields seem very low.

Table 1: round to 1 decimal

 Discussion and Conclusions from the study are good and well presented.

Reviewer 2 Report

The experiment by Herrera et al. Aimed to evaluate the fermentation pattern and dry matter losses in corn silage inter-28 cropped with Marandu and Mombasa grasses in different sowing modalities through crop-livestock  integration. while the study was well designed, and the manuscript contents were well organized, I did identify some sentences that could be improved. 

Specific comments:

1. L155, 'and buffer capacity (BC)',  a comma should be added before 'and';

2. L158, 'proposed by [14]',  'indicated by [15]', the author name of the cited references is missed.

3. L167, L150, L259, and L261, also same question, the author name of the cited references is missed. I will stop these comments here. Check it and if needed, and make all these changes to the entire manuscript.

4. Discussion, the first sentence (L240-L242) should be included in a paragrah. Please rewrite the second sentence (L243-L246), ans it seems too long. 

5. L249, ‘Those authors’? who?

6. L249-L255, a too long sentence, please rewritte it. 

7, L274-L281,  a too long sentence, please rewritte it. 

8. Conclusions, the first sentence is too long. Please improve and shorten it. The last two sentences have to be separate paragraphs?
